# Usefulness of several factors and clinical scoring models in preoperative diagnosis of complicated appendicitis

Kenji Fujiwara[1,2]*, Atsushi Abe[1], Toshihiro Masatsugu[1], Tatsuya Hirano[1], Kiyohisa Hiraka[3], Masayuki Sada[1]

1 Department of Surgery, Sada Hospital, Fukuoka, Japan, 2 Department of Surgery and Oncology, Graduate School of Medical Sciences, Kyushu University, Fukuoka, Japan, 3 Department of Radiology, Sada Hospital, Fukuoka, Japan

* kengdom@surg1.med.kyushu-u.ac.jp

**Data Availability Statement:** Original full data cannot be shared publicly because of containing detailed surgical information like precise surgical

## Abstract

### Background

The preoperative distinction between uncomplicated and complicated appendicitis is important to determine the appropriate treatments, such as antibiotics, surgery, or interval appendectomy. Computed tomography (CT) plays an important role; however, combining clinical and imaging factors may make preoperative evaluation more reliable. This study evaluated and analyzed cases and the usefulness of several preoperative factors and clinical scoring models to detect complicated appendicitis.

### Methods

A total of 203 patients preoperatively diagnosed with acute appendicitis at our facility were included. Complicated appendicitis was defined as appendicitis with gangrene, perforated appendix, and/or abscess formation. Preoperative factors were collected from published clinical scoring models; patient information, symptoms, signs, results of laboratory tests, and findings of CT. Factors were analyzed using a chi-squared test and the Mann-Whitney U test.

### Results

The preoperative factors were compared between 151 uncomplicated and 52 complicated appendicitis patients. The significant factors were age $\geq$40, duration of symptoms >24 hours, body temperature $\geq$37.3˚C, high levels of CRP, findings in CT scan (appendix diameter $\geq$10 mm, stranding of the adjacent fat, presence of fluid collection, and suspicion of abscess or perforation). We also evaluated the usefulness of clinical scoring models for the detection of complicated appendicitis and found the Appendicitis Inflammatory Response score and two prediction models (Atema score and Imaoka score) showed significance ($p <$ 0.05). High serum CRP level was significantly associated with complicated appendicitis ($p <$ 0.001), and the predicted existence rates of complicated appendicitis were 52.7% for serum CRP level $\geq$50mg/L, 74.4% for $\geq$100mg/L, and 82.6% for $\geq$150mg/L.

date. This information may indicate the patients' information. Data after being anonymized are available from the Sada Hospital Institutional Review Board (contact via e-mail; info@sada.or.jp) for researchers who meet the criteria for access to confidential data.

**Funding:** The author(s) received no specific funding for this work.

**Competing interests:** The authors have declared that no competing interests exist.

## Conclusion

The results demonstrated several preoperative factors and clinical scoring models to increase suspicion of complicated appendicitis. Specifically, high serum levels of CRP may be a useful factor in predicting complicated appendicitis prior to surgery when supported by clinical findings and imaging; however, further research is needed.

## Introduction

Acute appendicitis is a common affliction; however, the strategy for treating this common inflammatory condition has not been determined [1]. While emergency surgery is often performed for acute appendicitis in order to avoid progression of the condition [2], several studies have reported that antibiotics may treat uncomplicated appendicitis with high success rates of 88–94% [3, 4]. In addition, recent studies demonstrated that complicated appendicitis, defined as having a gangrenous appendix, perforated appendix, or periappendiceal abscess, could also be treated with antibiotics and surgical standby, contrary to the standard thought of complicated appendicitis as a typical candidate for emergency surgery [5, 6]. This method, interval appendectomy, might present fewer complications compared to emergency surgery [7]. Although the debate about whether interval appendectomy after non-operative management is necessary for complicated appendicitis continues, the distinction between uncomplicated and complicated appendicitides is important in deciding the strategy for treatment [8].

The preoperative distinction between uncomplicated and complicated appendicitides is difficult [8]. The diagnosis of acute appendicitis itself is challenging, and studies reported that normal appendixes were found in 5% of patients who had been diagnosed with acute appendicitis using imaging prior to surgery [1, 9]. Salminen et al. reported 1.5% of patients preoperatively diagnosed as having uncomplicated appendicitis, even with confirmation using computed tomography (CT), were then diagnosed as having complicated appendicitis during surgery; it is worth noting that this study excluded many patients (61.6% of all patients) for several factors like the presence of appendicolith, age, evidence of peritonitis, and so forth [3]. While CT plays an important role in detecting complicated appendicitis [1, 8], Atema et al. reported that combining clinical and imaging features were essential for correctly identifying uncomplicated appendicitis as well [10]. From this, it is clear that combining several factors, including imaging and clinical features, is important for the preoperative distinction between uncomplicated and complicated appendicitides.

Several studies have reported on preoperative factors and clinical scoring models used in the diagnosis of acute appendicitis and the prediction of severity of the condition [1, 8, 10–14]. However, each model proposes various factors and different thresholds. We would like to know definitive factors or scoring models to suspect complicated appendicitis preoperatively. Therefore, for this study, we evaluated the usefulness of those factors and scoring models in detecting complicated appendicitis by using our data.

## Methods

### Patients' characteristics

We collected the data of patients who had undergone surgery at Sada Hospital, and who had been given a preoperative diagnosis of acute appendicitis, from November 2015 to August 2020. A total of 203 cases with pathological diagnoses and findings of CT scan were included,

**Table 1. The breakdown of the pathological diagnoses of 203 patients.**

| Pathological diagnosis | Cases (% of total) | Age range (median) (years) | Sex (male/female) | Patients with abscess or perforation (% of total) |
|---|---|---|---|---|
| Phlegmonous | 147 (72.4) | 11–84 (37) | 84/63 | 18 (12.3) |
| Gangrenous | 28 (13.8) | 14–75 (43.5) | 12/16 | 25 (89.3) |
| Minimal change | 11 (5.4) | 16–66 (26) | 5/6 | 1 (9.1) |
| Chronic appendicitis | 6 (3.0) | 19–56 (41) | 3/3 | 2 (33.3) |
| Acute diverticulitis | 5 (2.5) | 32–53 (39) | 5/0 | 2 (40.0) |
| Neoplasms* | 3 (1.5) | 31–82 (63) | 0/3 | 1 (33.3) |
| Granulomatous appendicitis | 2 (1.0) | 37–40 (38.5) | 1/1 | 0 |
| Fibrinous serositis | 1 (0.5) | 44 (44) | 1/0 | 0 |
| Total | 203 | 11–84 (38) | 111/92 | 49 (24.1) |

*Neoplasms include adenocarcinoma, mucinous cystic neoplasm, and microcarcinoid.

after excluding 5 cases of patients who underwent standby surgery after being treated with antibiotics. The breakdowns of pathological diagnoses and basic demographic information of the 203 patients are provided in Table 1.

## Data management

We defined complicated appendicitis, also called complex appendicitis, as appendicitis with gangrene, a perforated appendix, and/or appendicitis with abscess formation in accordance with the article by Bhangu A. et al. [1, 6]. For classifying the cases as either uncomplicated or complicated appendicitis, we utilized the pathological diagnoses provided by pathologists in the case files and referred to the surgical records to determine the existence of abscess and perforation. Any appendicitides fitting the definition of complicated appendicitis were assigned to the complicated group, and all others were assigned to the uncomplicated group.

Several studies were reviewed, and their preoperative factors used in the diagnosis of acute appendicitis and the prediction of severity of the condition were considered [1, 8, 11, 12]. Especially, we mainly collected the factors using for scoring in three clinical risk score models for the diagnosis of acute appendicitis and two scoring models for the prediction of complicated appendicitis: Alvarado score, Appendicitis Inflammatory Response (AIR) score, Adult Appendicitis Score (AAS), the prediction model by Atema et al., and the prediction model by Imaoka et al. [9, 10, 13, 15]. The preoperative factors determined for use in this study were 1) patient information: age, sex, duration of symptoms (from the appearance of symptoms till visiting hospital firstly), 2) symptoms: nausea, vomiting, symptoms of anorexia, 3) signs: body temperature, pain in the right lower quadrant, rebound tenderness or muscular defense, 4) laboratory tests: level of C-reactive protein (CRP), white blood cell (WBC) count, leucocytosis shift, polymorphonuclear leucocytes, 5) findings of CT: appendix diameter, adjacent fat stranding, presence of fluid collection, suspicion of abscess or perforation, and suspicion of appendicolith. The findings of CT were determined by radiologists and surgeons according to Radiopedia (http://radiopedia.org/) or published articles [16, 17]. In our clinical records, some information like symptoms or leukocytosis shift was not recorded or analyzed for some patients, so some tables in this manuscript show different total numbers.

## Ethics statement

Sada Hospital has its own Institutional Review Board (IRB) that reviews all studies performed in the hospital. This IRB approved the use of the hospital database for research purposes and

waived the requirement for informed consent (IRB approval number: S200911-1). All data were fully anonymized before being assessed.

**Statistical analysis.** The preoperative factors and the scoring models were analyzed using a chi-squared test. The level of CRP was also studied with the Mann-Whitney U test. Statistical analysis was performed using JMP Pro 15.1.0 (SAS Institute Inc., Cary, NC, USA). A *p*-value of <0.05 was considered statistically significant.

## Results

### Usefulness of preoperative factors to predict complicated appendicitis

A total of 203 patients were classified as 151 with uncomplicated appendicitis (74.4%) and 52 with complicated appendicitis (25.6%). The 52 complicated cases contained 28 gangrenous appendicitis cases, 18 phlegmonous appendicitis cases, 2 chronic appendicitis cases, 2 acute diverticulitis cases, 1 minimal change case, and 1 mucinous cystic neoplasm and most of the cases had evidence of abscess/perforation except for 3 gangrenous appendicitis cases. We compared the relevant preoperative factors between the uncomplicated and complicated groups; these results are summarized in Table 2. The factors that showed significantly higher incidence in patients finally diagnosed with complicated appendicitis compared to those finally diagnosed with uncomplicated appendicitis were: aged ≥40 years (66.7% and 41.1%, respectively; *p* = 0.002), duration of symptoms ≥24 hours (67.3% and 29.8%, respectively; *p* < 0.001), body temperature ≥37.3°C (71.2% and 36.4%, respectively; *p* < 0.001), serum CRP level ≥50mg/L (76.5% and 23.2%, respectively; *p* < 0.001), appendix diameter ≥10 mm (90.4% and 63.6%, respectively; *p* < 0.001), stranding of the adjacent fat (96.2% and 66.2%, respectively; *p* < 0.001), presence of fluid collection (69.2% and 11.3%, respectively; *p* < 0.001), and suspicion of abscess or perforation (40.4% and 1.3%, respectively; *p* < 0.001).

### Findings of CT imaging with complicated appendicitis

As mentioned, CT is frequently used for the diagnosis of acute appendicitis and for evaluating the severity of appendicitis [8]. In our data, the CT finding of a suspicious abscess (such as fluid collection with rim enhancement) or perforation (e.g., the existence of free air outside of the gut) was not highly found in complicated appendicitis (21 of 52 cases or 38.9%, Table 2). During surgery, 28 cases of complicated appendicitis revealed infected fluid collected around the appendix or perforation of appendicitis; these patients did not demonstrate findings to cause suspicion of an abscess or perforation (Fig 1). Most of the complicated appendicitis patients showed appendix diameter 10 mm or larger (90.4%) and stranding of the adjacent fat (96.2%); these findings were also frequent for uncomplicated appendicitis (63.6% and 66.2%, respectively). The presence of fluid collection during CT might well indicate complicated appendicitis (36 of 52 cases; sensitivity 69.2%); 17 cases of uncomplicated appendicitis show fluid collection (17 of 151 cases; false positive was 11.3%). From these results, CT finding is useful but not perfect to distinguish preoperatively between uncomplicated and complicated appendicitides. We may combine other factors to increase the accuracy of preoperative distinction [10].

### Comparison between clinical scoring models regarding preoperative prediction of complicated appendicitis

We hypothesized that the clinical risk score models used for the accurate diagnosis of acute appendicitis might also be beneficial for the prediction of complicated appendicitis. Three clinical risk score models of acute appendicitis were chosen: Alvarado score, Appendicitis

**Table 2. Comparison of preoperative factors between uncomplicated and complicated appendicitis groups.**

| | | Uncomplicated appendicitis | Complicated appendicitis | *p*-value |
|---|---|---|---|---|
| **Patient information** | | | | |
| Age (years) | ≥40 | 62 | 34 | 0.002 |
| | <40 | 89 | 17 | |
| Duration of symptoms (hours) | ≥24 | 45 | 35 | <0.001 |
| | <24 | 106 | 17 | |
| **Symptoms** | | | | |
| Nausea/vomiting | Yes | 57 | 21 | 0.736 |
| | No | 94 | 31 | |
| Anorexia | Yes | 58 | 25 | 0.235 |
| | No | 92 | 27 | |
| **Signs** | | | | |
| Body temperature (˚C) | ≥37.3 | 55 | 37 | <0.001 |
| | <37.3 | 96 | 15 | |
| Pain in right lower quadrant | Yes | 142 | 50 | 0.314 |
| | No | 8 | 1 | |
| Rebound tenderness | Yes | 44 | 16 | 0.370 |
| | No | 74 | 19 | |
| **Laboratory tests** | | | | |
| Level of CRP (mg/L) | ≥50 | 35 | 39 | <0.001 |
| | <50 | 116 | 12 | |
| Count of WBCs (K/uL) | ≥15.0 | 47 | 15 | 0.912 |
| | 10.0–14.9 | 79 | 29 | |
| | <10.0 | 25 | 8 | |
| **Findings of CT** | | | | |
| Appendix diameter | ≥10 mm | 96 | 47 | <0.001 |
| | <10 mm | 55 | 5 | |
| Stranding of the adjacent fat | Yes | 100 | 50 | <0.001 |
| | No | 51 | 2 | |
| Presence of fluid collection | Yes | 17 | 36 | <0.001 |
| | No | 134 | 16 | |
| Suspicion of abscess/perforation | Yes | 2 | 21 | <0.001 |
| | No | 149 | 31 | |
| Presence of appendicolith | Yes | 48 | 19 | 0.530 |
| | No | 103 | 33 | |

CRP, C-reactive protein; WBC, white blood cell/leukocyte.

Inflammatory Response (AIR) score, and Adult Appendicitis Score (AAS) [8–10, 12, 13, 15]. The numbers of patients between uncomplicated and complicated appendicitis groups were contrasted by determining the risk scores of each patient using each model and then comparing the results (Table 3). Only AIR scores showed significance between the score and the existence of complicated appendicitis (*p* = 0.026). In addition, two scoring models for the prediction of complicated appendicitis were also tested. Atema score and Imaoka score both showed significance between the score and the existence of complicated appendicitis (*p* < 0.001).

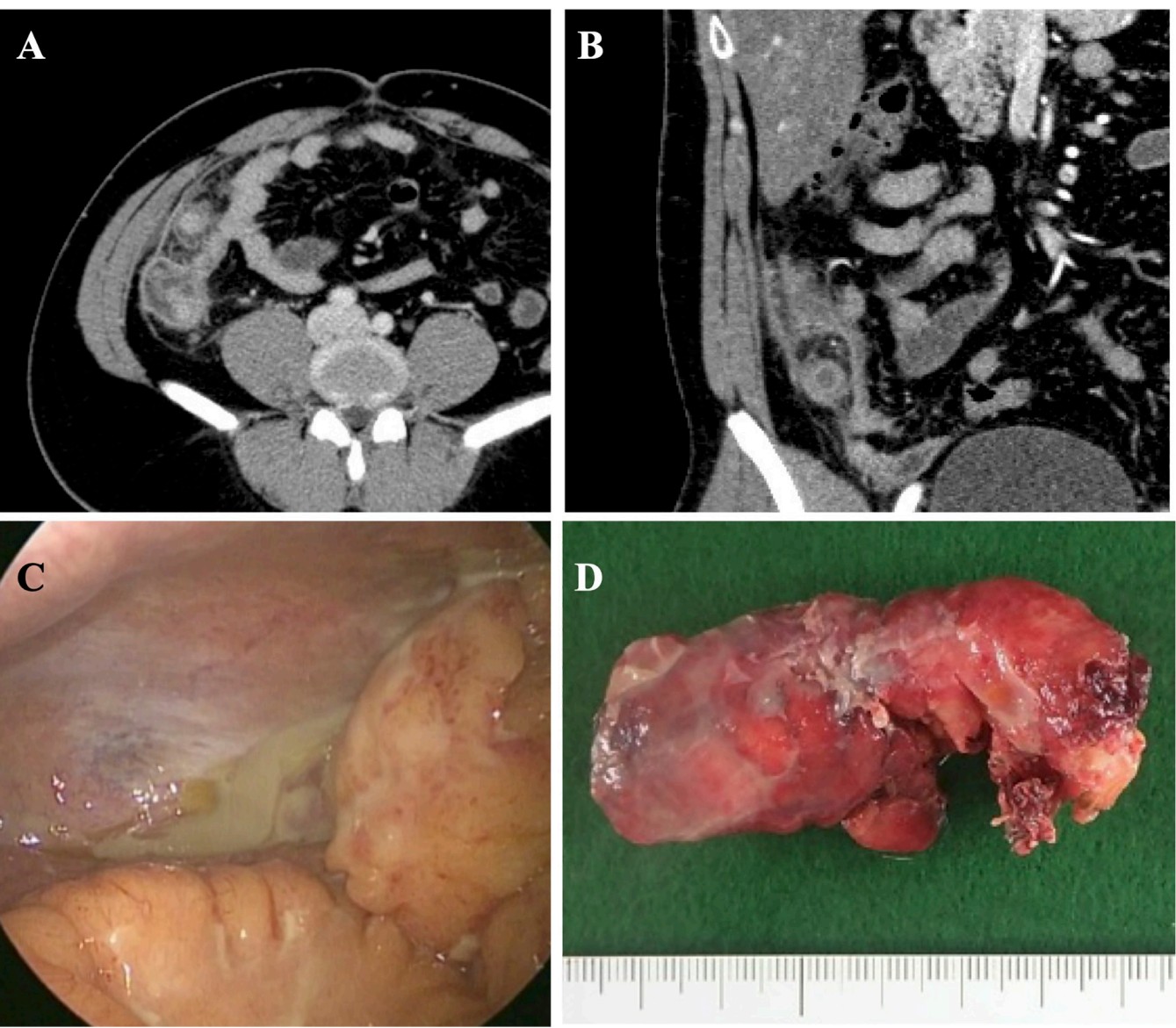

**Fig 1. A case of complicated appendicitis.** (A, B) Computed tomography images (A, transverse plane; B, coronal plane). A swollen appendix, stranding of the adjacent fat, and fluid collection are found; no obvious abscess is detected. (C) View during abdominal laparoscopy. Collection of infected fluid is found on the right side of abdomen. (D) Macroscopic image of the resected appendix. The appendix did not have signs of necrosis or perforation and was diagnosed as phlegmonous appendicitis.

### Relationship between the level of CRP and the incidence of complicated appendicitis

Our analysis found serum level of CRP, defined as $\geq$ 50 mg/L, was significantly associated with complicated appendicitis ($p < 0.001$, Table 2), and three clinical scoring models that showed significance for predicting complicated appendicitis used level of CRP for their scoring. We compared the serum levels of CRP of the patients between uncomplicated and complicated appendicitis groups. The results showed that CRP level was significantly higher in the complicated appendicitis group compared by uncomplicated appendicitis group ($p < 0.001$, Fig 2A). We analyzed the sensitivity and specificity of the serum CRP levels by setting each

**Table 3. Comparison between clinical scoring models for preoperative severity between uncomplicated and complicated appendicitis.**

| | Score | Uncomplicated appendicitis (% of total) | Complicated appendicitis (% of total) | *p*-value |
|---|---|---|---|---|
| **Clinical risk scoring models for suspected acute appendicitis** | | | | |
| Alvarado score | 0–4 (Low risk of acute appendicitis) | 12 (13.8) | 1 (4.5) | 0.274 |
| | 5–6 (Intermediate risk) | 23 (26.4) | 4 (18.2) | |
| | 7–10 (High risk) | 52 (59.8) | 17 (77.3) | |
| AIR score | 0–4 (Low risk) | 23 (21.5) | 1 (3.8) | 0.026 |
| | 5–8 (Intermediate risk) | 79 (73.8) | 21 (80.8) | |
| | 9–12 (High risk) | 5 (4.7) | 4 (15.4) | |
| AAS | 0–10 (Low risk) | 12 (12.0) | 1 (4.5) | 0.248 |
| | 11–15 (Intermediate risk) | 63 (63.0) | 12 (54.5) | |
| | 16+ (High risk) | 25 (25.0) | 9 (40.9) | |
| **Scoring models to predict complicated appendicitis** | | | | |
| Atema score | 0–6 (Low probability of complicated appendicitis) | 124 (82.7) | 5 (9.8) | <0.001 |
| | 7+ (High probability) | 26 (17.3) | 46 (90.2) | |
| Imaoka score | 0 (Low probability) | 72 (47.7) | 1 (2.0) | <0.001 |
| | 1–3 (High probability) | 79 (52.3) | 49 (98.0) | |

AIR, Appendicitis Inflammatory Response; AAS, Adult Appendicitis Score.

cut-off value (Table 4) and created a receiver operating characteristic curve (Fig 2B). The area under the curve was 0.843, and high serum CRP level was a significant indication factor for complicated appendicitis ($p < 0.001$). The predicted existence-rates (positive predictive values)

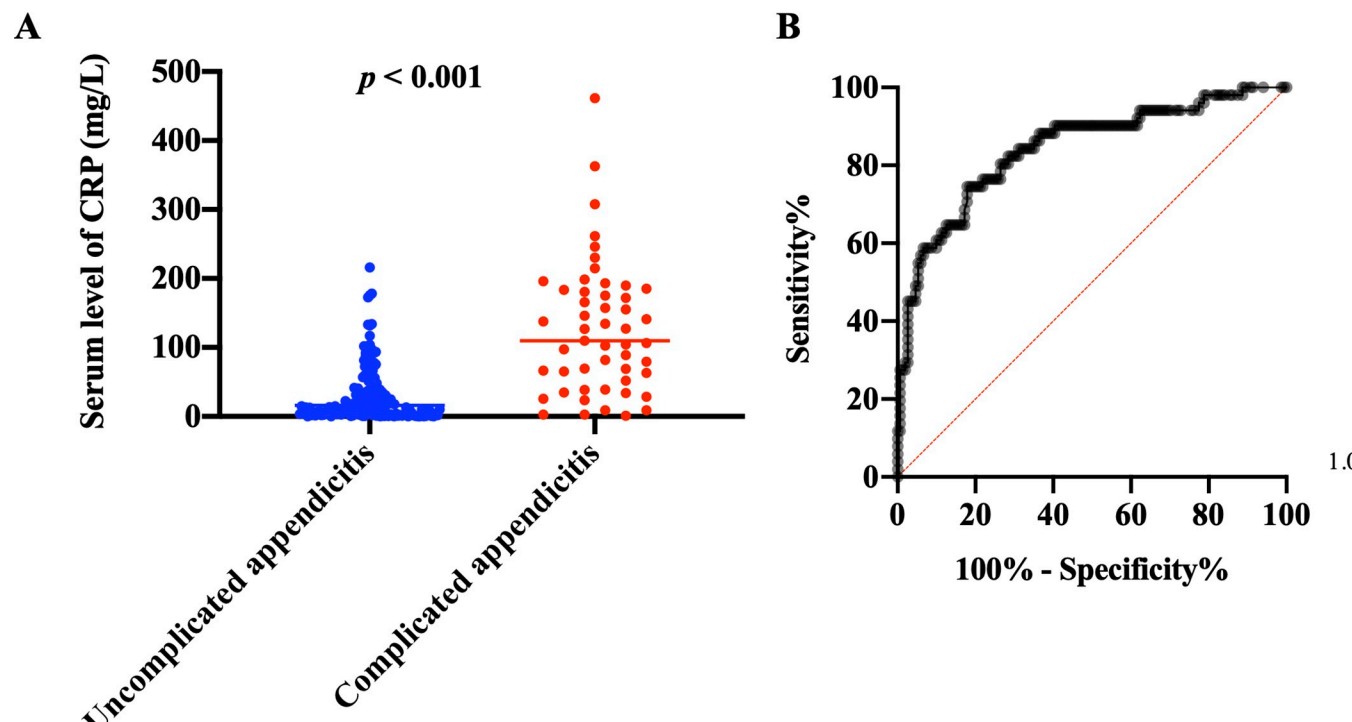

**Fig 2. The relationship between the level of C-reactive protein (CRP) and the existence of complicated appendicitis.** (A) Comparison of serum level of CRP between uncomplicated and complicated appendicitis. Bars show median values. (B) Receiver operating characteristic curve of the relationship between the level of CRP and the existence rates of complicated appendicitis. The area under the curve was 0.843.

**Table 4. Sensitivities, specificities, and predicted existence rates for cut-off values of serum level of C-reactive protein (CRP).**

| Cut-off value of serum level of CRP (mg/L) | Sensitivity (%) | Specificity (%) | Predicted existence-rate of complicated appendicitis (%; positive predictive value) |
|---|---|---|---|
| ≥10 | 90.2 | 39.7 | 33.6 |
| ≥50 | 76.5 | 76.8 | 52.7 |
| ≥100 | 56.9 | 93.4 | 74.4 |
| ≥150 | 37.3 | 97.4 | 82.6 |

of complicated appendicitis were 52.7% for serum CRP level ≥50mg/L, 74.4% for ≥100mg/L, and 82.6% for ≥150mg/L.

## Discussion

The diagnosis and evaluation of the severity of acute appendicitis remain challenging, even though surgery has been frequently performed to treat this common condition all over the world [1]. While CT is usually used for definitive evaluation, our data demonstrated that many complicated appendicitis cases (59.6%) did not show the expected images, such as fluid collection with rim enhancement or free air in the abdomen. The stranding of adjacent fat and swelling of the appendix were found in most of the complicated appendicitis cases. However, these findings were also frequently shown in uncomplicated appendicitis (63.6–66.2%), so specificity is not high. The presence of fluid collection seemed a reasonable factor to suspect complicated appendicitis due to the balance of sensitivity (69.2%) and specificity (88.7%). Two scoring models (Atema score and Imaoka score) for the prediction of complicated appendicitis both contained the presence of fluid collection for scoring and also both models recommended combining other factors such as serum level of CRP [10, 14].

CRP may be one useful indicator of complicated appendicitis due to simplicity and objectivity. Comparison among three well-known clinical risk score models showed that only the AIR score demonstrated significance. This might be because the AIR score added a progressively higher score for higher serum levels of CRP. The AAS also used CRP in scoring; however, this scoring model was applied in an inconsistent manner and did not always show the highest score for patients with the highest levels of CRP [13], and the Alvarado score did not use the serum CRP level for its scoring [1]. For the ideal threshold of serum CRP level, Atema score and Imaoka score presented a serum level of 47–50 mg/L. By using our data, both models showed significance for the prediction of complicated appendicitis so we thought the threshold of serum CRP level from the two models is appropriate for the scoring models of the combination of several factors.

This study has the limitation of sample size. Given the significant $p$-values in our data, we believe that our results remain relevant; however, we acknowledge that further research with larger numbers of cases is required to detect independent factors by multivariate analysis. In addition, the comparison among studies about complicated appendicitis had several limitations. First, many studies used a different definition for complicated appendicitis, such as the presence of an appendicolith, periappendiceal phlegmon, or peritonitis [3, 4, 10, 13, 15]. For comparing the preoperative factors and scoring models with consistent definition, we chose the definition of Bhangu et al. because of its simplicity and objectivity: appendicitis with gangrene, perforated appendix, and/or abscess formation [1, 6]. Due to many definitions of complicated appendicitis, it is important to be careful when comparing studies about complicated appendicitis in order to avoid confusion. Secondly, the availability of imaging like CT is different among facilities although in this study CT was performed for most cases [1]. Atema score also showed the scoring models by using ultrasounds and we can utilize it [10]; however, the reliability of the results by ultrasounds depends on the technique of the operator.

In conclusion, the results showed several preoperative factors and clinical scoring models combining several factors were useful to detect complicated appendicitis. We propose that CRP may be a useful factor in predicting complicated appendicitis when supported by clinical findings and imaging, and look forward to further research of this factor.

## Acknowledgments

We want to thank Dr. Akinori Iwashita for the pathological diagnoses and advice given in these acute appendicitis cases. We would also like to thank all the medical staff of Sada Hospital who contributed to the treatment of acute appendicitis and aided in collecting the data for our research.

## Author Contributions

**Conceptualization:** Kenji Fujiwara.

**Data curation:** Kenji Fujiwara, Atsushi Abe, Toshihiro Masatsugu, Tatsuya Hirano, Kiyohisa Hiraka, Masayuki Sada.

**Formal analysis:** Kenji Fujiwara.

**Methodology:** Kenji Fujiwara, Kiyohisa Hiraka.

**Project administration:** Kenji Fujiwara.

**Supervision:** Masayuki Sada.

**Writing – original draft:** Kenji Fujiwara.

**Writing – review & editing:** Atsushi Abe, Toshihiro Masatsugu, Tatsuya Hirano, Kiyohisa Hiraka, Masayuki Sada.

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
