## [Decision Letter · Decision Letter 0]

30 Apr 2021

PONE-D-21-12222

Usefulness of several factors and clinical scoring models in preoperative diagnosis of complicated appendicitis

PLOS ONE

Dear Dr. Fujiwara,

Thank you for submitting your manuscript to PLOS ONE. After careful consideration, we feel that it has merit but does not fully meet PLOS ONE’s publication criteria as it currently stands. Therefore, we invite you to submit a revised version of the manuscript that addresses the points raised during the review process.

Please revise accordingly.

We look forward to receiving your revised manuscript.

Kind regards,

Academic Editor

PLOS ONE

Journal Requirements:

Reviewers' comments:

Reviewer's Responses to Questions

**Comments to the Author**

1. Is the manuscript technically sound, and do the data support the conclusions?

Reviewer #1: Partly

Reviewer #2: No

2. Has the statistical analysis been performed appropriately and rigorously? 

Reviewer #1: Yes

Reviewer #2: I Don't Know

3. Have the authors made all data underlying the findings in their manuscript fully available?

Reviewer #1: Yes

Reviewer #2: Yes

4. Is the manuscript presented in an intelligible fashion and written in standard English?

Reviewer #1: No

Reviewer #2: Yes

5. Review Comments to the Author

Reviewer #1: This is an interesting topic to discuss, however, the case number was relatively small, especially the number of complicated appendicitis which was only 14.

Here, I have some concerns:

1. I agree with the idea that preoperative distinction between uncomplicated and complicated appendicitis is important in determining the appropriate treatments, such as antibiotics, surgery, or interval appendectomy. However, if not considering the suitableness of patients’ general condition such as underlying diseases and the severity of infection.

The logic of distinguishing complicated appendicitis from uncomplicated appendicitis is to make the best treatment decision for the patient instead of simply categorizing the “appendicitis.”

2. The laboratory and image criteria in differentiating complicated and uncomplicated appendicitis may not be representative enough.

For example, the abdominal CT scan findings can provide important clues of the inflammation severity, such as the amount of fluid accumulation as well as localization. Additionally, the area of adjacent fat stranding is a key finding of inflammation. The CT image grading for acute pancreatitis can be a model compared to your study by applying grade A-D to show the severities of inflammation. It would be informative if the authors can provide more details.

3. In figure 1B, the author presented an example of complicated appendicitis. However, the swollen appendix was isolated from the cecum and small intestines. It may not be considered as complicated appendicitis for experienced surgeons. In our experience, it won’t be too difficult when applying laparoscopic appendectomy. On the other hand, if the swollen appendix adhered to the adjacent structures or located retro-cecum or retro-ileum, we will be more cautious and consider it as complicated appendicitis.

Reviewer #2: The data are consistent with most of the notion. However, there are still several points need to be addressed in the manuscript. The main topic is not interesting and not sound in scientific research.

6. PLOS authors have the option to publish the peer review history of their article (what does this mean?). If published, this will include your full peer review and any attached files.

Reviewer #1: **Yes: **Guo-Shiou Liao

Reviewer #2: No

---

## [Author Response · Author response to Decision Letter 0]

12 Jun 2021

Reviewer #1: This is an interesting topic to discuss, however, the case number was relatively small, especially the number of complicated appendicitis which was only 14.

Author Response: Thank you for an excellent opinion. We agreed that the case number seemed small although the data showed significance. We decided to increase the number of cases, and eventually, we collected 203 appendicitides including 52 complicated appendicitis cases. After re-analyzing the data, we revised many tables and figures. The tendencies had not been changed despite drastic changes, so we kept the similar discussion and conclusion. We feel sorry for you for taking your time for the re-review of this drastic updating manuscript, but we strongly believe that our extended collection of the cases increased the reliability of our manuscript.

We summarized our revisions caused subsequently by the increase of total case number.

1. The sample number was changed, so the numbers of patients, percentages, and also p-values were revised. We used the function of the track change in the manuscript so you could find the revised points. All tables (Table 1-4) and Fig 2 were revised.

2. We removed the previous Table 3 (the list of 14 complicated appendicitis cases) because the list would become too huge with the increase of total numbers of complicated appendicitis. We arranged the part of “Findings of CT imaging with complicated appendicitis” in order to make the readers understand without the list of complicated appendicitis cases. 

By removing Table 3, previous Table 4 and 5 changed to new Table 3 and 4.

1. I agree with the idea that preoperative distinction between uncomplicated and complicated appendicitis is important in determining the appropriate treatments, such as antibiotics, surgery, or interval appendectomy. However, if not considering the suitableness of patients’ general condition such as underlying diseases and the severity of infection.

The logic of distinguishing complicated appendicitis from uncomplicated appendicitis is to make the best treatment decision for the patient instead of simply categorizing the “appendicitis.”

Author Response: Thank you for a great opinion. We completely agree with your statement that we should select the best treatment for the patients. The distinction between uncomplicated and complicated appendicitis should be for the best selection of the treatment. Actually, we wanted to warn the current tendencies to diagnose mainly with CT scans. Recently, many guidelines have been published for the treatment of appendicitis, and the distinction between uncomplicated and complicated appendicitis has been often used[1][2]. The distinction of uncomplicated or complicated appendicitis is convenient for the selection of treatment, but the definition of “complicated appendicitis” is not the same among the manuscripts[3][4][5][6][7]. Also, CT is frequently used for the diagnosis of complicated appendicitis, but we have often experienced perforation or abscess that was not diagnosed in CT scan. The diagnosis depending on CT scan may incorrectly define some severe cases like “complicated appendicitis” as un-severe cases like “uncomplicated appendicitis.” We would like to warn the problem of the diagnosis depending on mainly CT scan and emphasize the importance of the evaluation with other factors. 

From your suggestion, we thought our statement of the manuscript might not be clear, so we newly added the sentence at the end of the paragraph of “Findings of CT imaging with complicated appendicitis” on Page 11, “From these results, CT finding is useful but not perfect to distinguish preoperatively between uncomplicated and complicated appendicitides. We may combine other factors to increase the accuracy of preoperative distinction.”

2. The laboratory and image criteria in differentiating complicated and uncomplicated appendicitis may not be representative enough.

For example, the abdominal CT scan findings can provide important clues of the inflammation severity, such as the amount of fluid accumulation as well as localization. Additionally, the area of adjacent fat stranding is a key finding of inflammation. The CT image grading for acute pancreatitis can be a model compared to your study by applying grade A-D to show the severities of inflammation. It would be informative if the authors can provide more details.

Author Response: Thank you for your great opinion. On page 7, we added the sentence into the Method about how we diagnosed the findings of CT scan, “The findings of CT were determined by radiologists and surgeons according to Radiopedia (http://radiopedia.org/) or published articles.” I think it is a great idea to newly provide CT image grading for appendicitis, but we would like to it for the next project. In this manuscript, we tried to diagnose the findings with the usual and popular sense by referring to Radiopedia or published articles.

3. In figure 1B, the author presented an example of complicated appendicitis. However, the swollen appendix was isolated from the cecum and small intestines. It may not be considered as complicated appendicitis for experienced surgeons. In our experience, it won’t be too difficult when applying laparoscopic appendectomy. On the other hand, if the swollen appendix adhered to the adjacent structures or located retro-cecum or retro-ileum, we will be more cautious and consider it as complicated appendicitis.

Author Response: Thank you for a great question. We agree that some experienced surgeons would not consider this case as complicated appendicitis. And, other surgeons would think this diagnosis is complicated appendicitis. Such confusion about the diagnosis of severity was our motivation to start this research. We think the definition of complicated appendicitis should be objective but many manuscripts used each own definition. We chose Bhangu’s manuscript because they used a relatively objective definition of complicated appendicitis; gangrene, perforation, and/or abscess[2]. And we applied this definition to our experienced cases and compared several factors and risk score models. Actually, we did not find a perfect factor or perfect scoring model. Atema score seemed better than other factors due to the balance of sensitivity and specificity, but the calculation of their scoring model seemed complex[5]. We think that a combination of CT findings and a high serum level of CRP is easy and good to detect severe cases but further research and discussion are necessary. We added the predicted existence rate of complicated appendicitis in Table 4. We wish this table would support surgeons to explain the patients the severity of their appendicitis and support choosing the treatment.

Additional major revisions

1. We found low serum levels of CRP for some complicated appendicitis cases by increasing the total numbers of patients although we described all serum levels of CRP were above 50 mg/L for complicated appendicitis patients before. In the abstract and the end of the discussion, we changed the definition of CRP as “a useful factor” from “a strong factor” even though CRP still showed a high significance for the prediction of complicated appendicitis (p<0.0001).

2. We changed Fig 2A from bar graph to dot-plot graph. We thought the dot-plot style is more appropriate for our data because the serum level of CRP was sparse, and the dot-plot can also show a very high-level value in one figure.

Minor revision.

1. In the Method, we did not describe clearly that we did not include the patients not having CT findings in the previous manuscript. Therefore, we added the sentence “A total of 203 cases with pathological diagnoses and findings of CT scan were included” in order to make it clear. We excluded seven patients diagnosed with only ultrasounds, and we think the rejection of those seven patients did not affect the whole analysis.

2. Some data showed different total numbers due to incomplete records of symptoms or un-enforcement of some laboratory tests. On pages 7-8, we added this sentence at the end of the section of Data management, “In our clinical records, some information like symptoms or leukocytosis shift was not recorded or analyzed for some patients, so some tables in this manuscript show different total numbers.”

3. In the part of “Statistical analysis” of the Method, we removed the sentence “graphic presentations were performed using JMP Pro 15.1.0” because we also used other software in order to remove the gray background of the ROC curve made by JMP Pro.

 

Reviewer #2: The data are consistent with most of the notion. However, there are still several points need to be addressed in the manuscript. The main topic is not interesting and not sound in scientific research.

Author Response: Thank you for a great opinion. We seriously thought about your opinion, and we agreed that our manuscript was not at the level of scientific research and not interesting. We discussed how to improve this manuscript and decided to increase the number of cases to make the manuscript more reliable. Eventually, we collected a total of 203 cases, including 52 complicated appendicitis cases. The tendencies were not changed, and p-values became lower. We believe the increase of case number makes our manuscript more reliable.

Also, we thought the message of our manuscript was not clear. The motivation of this manuscript was the confusion of the research of appendicitis. Many manuscripts used different definitions of complicated appendicitis [3][4][5][6][7]. The dramatic change of the treatment for appendicitis has been progressed by introducing interval appendectomy[8]. On the other hand, we thought that the precise preoperative diagnoses of complicated appendicitis is difficult, and CT scan is not a perfect tool to detect complicated appendicitis. Therefore, we would like to evaluate the current useful factors to use the preoperative diagnosis of acute appendicitis by using the consistent definition of complicated appendicitis by applying it to our data. We used Bhangu’s criteria because their definition was simple and relatively objective; gangrene, perforation, and/or abscess[2]. In addition, on page 12, we added the sentence of our opinion in order to make our statement clear in the result section, “From these results, CT finding is useful but not perfect to distinguish preoperatively between uncomplicated and complicated appendicitides. We may combine other factors to increase the accuracy of preoperative distinction.” In addition, we resummarize the manuscript by deleting one table and delete some redundant sentences. We tried to make our manuscript more straightforward to tell our statement.

We summarize our revisions.

1. The sample number was changed, so the numbers of patients, percentages, and also p-values were revised. We used the function of Track change in the manuscript so you could find the revised points. All tables (Table 1-4) and Fig 2 were revised.

2. We removed the previous Table 3 (the list of 14 complicated appendicitis cases) because the list would become too huge with the increase of total numbers of complicated appendicitis. We arranged the part of “Findings of CT imaging with complicated appendicitis”in order to make the readers understand without the list of complicated appendicitis cases. 

By removing Table 3, previous Table 4 and 5 changed to new Table 3 and 4.

3. We found low serum levels of CRP for some complicated appendicitis cases by increasing the total numbers of patients although we described all serum levels of CRP were above 50 mg/L for complicated appendicitis patients before. In the abstract and the end of discussion, we changed the definition of CRP as “a useful factor” from “a strong factor” even though CRP still showed a high significance for the prediction of complicated appendicitis (p<0.0001).

4. We changed Fig 2A from bar graph to dot-plot graph. We thought the dot-plot style is more appropriate for our data because the serum level of CRP was sparse, and the dot-plot can also show a very high-level value in one figure.

Minor revision.

1. In the Method, we did not describe clearly that we did not include the patients not having CT findings in the previous manuscript. Therefore, we added the sentence “A total of 203 cases with pathological diagnoses and findings of CT scan were included” in order to make it clear. We excluded seven patients diagnosed with only ultrasounds, and we think the rejection of those seven patients did not affect the whole analysis.

2. Some data showed different total numbers due to incomplete records of symptoms or un-enforcement of some laboratory tests. On pages 7-8, we added this sentence at the end of the section of Data management, “In our clinical records, some information like symptoms or leukocytosis shift was not recorded or analyzed for some patients, so some tables in this manuscript show different total numbers.”

3. In the part of “Statistical analysis” of the Method, we removed the sentence “graphic presentations were performed using JMP Pro 15.1.0” because we also used other software in order to remove the gray background of the ROC curve made by JMP Pro.

References

1. Di Saverio S, Podda M, De Simone B, Ceresoli M, Augustin G, Gori A, et al. Diagnosis and treatment of acute appendicitis: 2020 update of the WSES Jerusalem guidelines. World J Emerg Surg. 2020;15: 27. 

2. Bhangu A, Søreide K, Di Saverio S, Assarsson JH, Drake FT. Acute appendicitis: modern understanding of pathogenesis, diagnosis, and management. Lancet (London, England). 2015;386: 1278–1287. 

3. Salminen P, Paajanen H, Rautio T, Nordström P, Aarnio M, Rantanen T, et al. Antibiotic Therapy vs Appendectomy for Treatment of Uncomplicated Acute Appendicitis: The APPAC Randomized Clinical Trial. JAMA. 2015;313: 2340–2348. 

4. Vons C, Barry C, Maitre S, Pautrat K, Leconte M, Costaglioli B, et al. Amoxicillin plus clavulanic acid versus appendicectomy for treatment of acute uncomplicated appendicitis: an open-label, non-inferiority, randomised controlled trial. Lancet (London, England). 2011;377: 1573–1579. 

5. Atema JJ, van Rossem CC, Leeuwenburgh MM, Stoker J, Boermeester MA. Scoring system to distinguish uncomplicated from complicated acute appendicitis. Br J Surg. 2015;102: 979–990. 

6. Sammalkorpi HE, Mentula P, Savolainen H, Leppäniemi A. The Introduction of Adult Appendicitis Score Reduced Negative Appendectomy Rate. Scand J Surg. 2017;106: 196–201. 

7. Gorter RR, Eker HH, Gorter-Stam MAW, Abis GSA, Acharya A, Ankersmit M, et al. Diagnosis and management of acute appendicitis. EAES consensus development conference 2015. Surg Endosc. 2016;30: 4668–4690. 

8. Andersson RE, Petzold MG. Nonsurgical treatment of appendiceal abscess or phlegmon: a systematic review and meta-analysis. Ann Surg. 2007;246: 741–748.

---

## [Decision Letter · Decision Letter 1]

25 Jun 2021

PONE-D-21-12222R1

Usefulness of several factors and clinical scoring models in preoperative diagnosis of complicated appendicitis

PLOS ONE

Dear Dr. Fujiwara,

Thank you for submitting your manuscript to PLOS ONE. After careful consideration, we feel that it has merit but does not fully meet PLOS ONE’s publication criteria as it currently stands. Therefore, we invite you to submit a revised version of the manuscript that addresses the points raised during the review process.

Please revise accordingly.

We look forward to receiving your revised manuscript.

Kind regards,

Academic Editor

PLOS ONE

Journal Requirements:

Reviewers' comments:

Reviewer's Responses to Questions

**Comments to the Author**

1. If the authors have adequately addressed your comments raised in a previous round of review and you feel that this manuscript is now acceptable for publication, you may indicate that here to bypass the “Comments to the Author” section, enter your conflict of interest statement in the “Confidential to Editor” section, and submit your "Accept" recommendation.

Reviewer #1: All comments have been addressed

Reviewer #3: All comments have been addressed

2. Is the manuscript technically sound, and do the data support the conclusions?

Reviewer #1: Partly

Reviewer #3: Yes

3. Has the statistical analysis been performed appropriately and rigorously? 

Reviewer #1: Yes

Reviewer #3: Yes

4. Have the authors made all data underlying the findings in their manuscript fully available?

Reviewer #1: Yes

Reviewer #3: Yes

5. Is the manuscript presented in an intelligible fashion and written in standard English?

Reviewer #1: Yes

Reviewer #3: Yes

6. Review Comments to the Author

Reviewer #1: For diagnosing acute appendicitis, both clinical and laboratory findings individually do not suffice, combined with

standard imaging increases the diagnostic power for acute appendicitis. Incorporating imaging features in clinical scoring models may provide better differentiation between uncomplicated and complicated appendicitis. Optimizing patient selection for treatment of appendicitis resulting in better treatment outcomes. Too emphasize the CRP value may misdiagnose with other abdominal diseases.

Reviewer #3: I have read with interest this submission. The authors have for the most part revised and addressed the recommendations of the very thorough review performed by the 1st round of reviewers. As I read the manuscript, I have only identified a minor issue which I think need to be addressed.

As it is known, CRP is a nonspecific acute phase reactant and may increase for many reasons. It would be a overestimated statement to say that CRP alone is a useful factor in predicting complicated appendicitis. In the discussion, it would be appropriate to edit the statement as "may be useful when supported by clinical findings and imaging".

7. PLOS authors have the option to publish the peer review history of their article (what does this mean?). If published, this will include your full peer review and any attached files.

Reviewer #1: No

Reviewer #3: No

---

## [Author Response · Author response to Decision Letter 1]

29 Jun 2021

PONE-D-21-12222

Usefulness of several factors and clinical scoring models in preoperative diagnosis of complicated appendicitis

PLOS ONE

June 29, 2021

Dear Editors and Reviewers,

We are very pleased to re-submit our manuscript, entitled “Usefulness of several factors and clinical scoring models in preoperative diagnosis of complicated appendicitis” by Fujiwara et al. for your consideration for publication in PLOS ONE.

Thank you for taking the time to review our revised manuscript. We believe our manuscript is now fit for the criteria for the journal. Below, we have listed each comment, along with our subsequent revisions and responses.

Sincerely,

Kenji Fujiwara, M.D., Ph.D.

 

Reviewer #1: For diagnosing acute appendicitis, both clinical and laboratory findings individually do not suffice, combined with standard imaging increases the diagnostic power for acute appendicitis. Incorporating imaging features in clinical scoring models may provide better differentiation between uncomplicated and complicated appendicitis. Optimizing patient selection for treatment of appendicitis resulting in better treatment outcomes. To emphasize the CRP value may misdiagnose with other abdominal diseases.

Author Response: Thank you for the thoughtful suggestion. We agreed that we exaggerated the importance of CRP too much, and this may mislead the readers. We arranged some sentences to indicate that CRP may be useful when supported by clinical findings and imaging. 

We summarized the changes below.

1. We added the sentence “when supported by clinical findings and imaging” in the abstract (Page 3): Specifically, high serum levels of CRP may be a useful factor in predicting complicated appendicitis prior to surgery when supported by clinical findings and imaging; however, further research is needed.

2. We deleted the sentences “Great benefit of standard blood draws is easier and less invasive than imaging like CT scan (Page 16; Line 240-241)” because this sentence may mislead readers to think that CRP can substitute the imaging. 

3. We added the sentence “when supported by clinical findings and imaging” in conclusion (Page 17): We propose that CRP may be a useful factor in predicting complicated appendicitis when supported by clinical findings and imaging, and look forward to further research of this factor.

 

Reviewer #3: I have read with interest this submission. The authors have for the most part revised and addressed the recommendations of the very thorough review performed by the 1st round of reviewers. As I read the manuscript, I have only identified a minor issue which I think need to be addressed.

As it is known, CRP is a nonspecific acute phase reactant and may increase for many reasons. It would be a overestimated statement to say that CRP alone is a useful factor in predicting complicated appendicitis. In the discussion, it would be appropriate to edit the statement as "may be useful when supported by clinical findings and imaging".

Author Response: Thank you for the thoughtful suggestion. We agreed that we exaggerated the importance of CRP too much, and this may mislead the readers. We arranged some sentences to indicate that CRP may be useful when supported by clinical findings and imaging. 

We summarized the changes below.

1. We added the sentence “when supported by clinical findings and imaging” in the abstract (Page 3): Specifically, high serum levels of CRP may be a useful factor in predicting complicated appendicitis prior to surgery when supported by clinical findings and imaging; however, further research is needed.

2. We deleted the sentences “Great benefit of standard blood draws is easier and less invasive than imaging like CT scan (Page 16; Line 240-241)” because this sentence may mislead readers to think that CRP can substitute the imaging. 

3. We added the sentence “when supported by clinical findings and imaging” in conclusion (Page 17): We propose that CRP may be a useful factor in predicting complicated appendicitis when supported by clinical findings and imaging, and look forward to further research of this factor.

---

## [Decision Letter · Decision Letter 2]

13 Jul 2021

Usefulness of several factors and clinical scoring models in preoperative diagnosis of complicated appendicitis

PONE-D-21-12222R2

Dear Dr. Fujiwara,

We’re pleased to inform you that your manuscript has been judged scientifically suitable for publication and will be formally accepted for publication once it meets all outstanding technical requirements.

Kind regards,

Academic Editor

PLOS ONE

Additional Editor Comments (optional):

Reviewers' comments:

Reviewer's Responses to Questions

**Comments to the Author**

1. If the authors have adequately addressed your comments raised in a previous round of review and you feel that this manuscript is now acceptable for publication, you may indicate that here to bypass the “Comments to the Author” section, enter your conflict of interest statement in the “Confidential to Editor” section, and submit your "Accept" recommendation.

Reviewer #1: All comments have been addressed

Reviewer #3: All comments have been addressed

2. Is the manuscript technically sound, and do the data support the conclusions?

Reviewer #1: Yes

Reviewer #3: Yes

3. Has the statistical analysis been performed appropriately and rigorously? 

Reviewer #1: Yes

Reviewer #3: Yes

4. Have the authors made all data underlying the findings in their manuscript fully available?

Reviewer #1: Yes

Reviewer #3: Yes

5. Is the manuscript presented in an intelligible fashion and written in standard English?

Reviewer #1: Yes

Reviewer #3: Yes

6. Review Comments to the Author

Reviewer #1: (No Response)

Reviewer #3: The authors have made all the corrections addressed by the reviewers. With these contributions, the study has become more valuable and ready for publication.

7. PLOS authors have the option to publish the peer review history of their article (what does this mean?). If published, this will include your full peer review and any attached files.

Reviewer #1: No

Reviewer #3: No

---

## [Editor Report · Acceptance letter]

19 Jul 2021

PONE-D-21-12222R2 

Usefulness of several factors and clinical scoring models in preoperative diagnosis of complicated appendicitis 

Dear Dr. Fujiwara:

I'm pleased to inform you that your manuscript has been deemed suitable for publication in PLOS ONE. Congratulations! Your manuscript is now with our production department. 

Kind regards, 

on behalf of

Dr. Robert Jeenchen Chen 

Academic Editor

PLOS ONE